# New Complexity Results for Structurally Restricted Numeric Planning

**Alexander Shleyfman[1], Daniel Gnad[2], Peter Jonsson[2]**

[1]The Faculty of Industrial Engineering and Management, Technion, Haifa, Israel
[2]Department of Computer and Information Science, Linköping University, Linköping, Sweden
shleyfman.alexander@gmail.com, daniel.gnad@liu.se, peter.jonsson@liu.se

## Abstract

Numeric planning is known to be undecidable even under severe restrictions. Prior work investigated the decidability boundaries by restricting the expressiveness of the planning formalism in terms of the numeric functions allowed in conditions and effects. In this work, we fix one specific such formalism, *simple* numeric planning (SNP), which, while only allowing linear conditions and action effects to add constants, is still undecidable. We analyse the complexity of SNP by (1) restricting the number of numeric variables, and (2) restricting the causal structure. First, we concentrate on numeric planning with exactly one (numeric) variable. We present a pseudo-polynomial algorithm to solve such tasks, and show **NP**-hardness and **PSPACE**-membership for the corresponding decision problem. Second, we restrict the interaction between variables in terms of the causal graph. As our main result, we show that SNP with an arbitrary number of numeric causal-graph leaf variables is decidable, and lies in **PSPACE** if the propositional state space has fixed size.

## Introduction

In recent years, significant progress has been made towards the development of methods that solve planning tasks with numeric variables (Hoffmann 2002; Shin and Davis 2005; Gerevini, Saetti, and Serina 2008; Eyerich, Mattmüller, and Röger 2009; Coles et al. 2013; Scala et al. 2016; Illanes and McIlraith 2017; Li et al. 2018; Scala, Haslum, and Thiébaux 2016; Scala et al. 2017; Aldinger and Nebel 2017; Piacentini et al. 2018a,b; Kuroiwa et al. 2021). From a theoretical perspective, the success of these methods raises a conundrum, since even the most simple forms of numeric planning are known to be undecidable (Helmert 2002). Undecidability has been proven using planning instances with an arbitrary number of numeric variables and arbitrary causal dependencies between variables. What happens if we bound the number of numeric variables by a constant? What influence do causal dependencies have? We consider this a very interesting setting and make a step towards a better understanding of restricted sub-classes of numeric planning in this work.

We focus on the *simple numeric planning* (SNP) formalism (Hoffmann 2003; Scala et al. 2016). While it only allows linear expressions in the goal and action conditions, and action effects to add constants, it is still undecidable. Our starting point is a simple numeric task (SNT) with a *single* numeric variable $x$, which provides a platform to analyze

more interesting structures. We remark that even this simple problem is non-trivial, namely **NP**-hard (Thm. 4). The basis of our investigation is a normal form, similar to the domain simplification of Helmert (2002), which abstracts from irrelevant details. Our main tool for proving complexity results is Thm. 1, which concerns the structure of the state space of one-variable numeric tasks: for every solvable task $\Pi_x$ one can compute a bounded interval $I$ such that there exists a plan $\pi$ that stays inside $I$ during execution. To solve $\Pi_x$, it is thus sufficient to explore a finite part of the underlying infinite state space, which makes the planning problem decidable. Moreover, we prove a pseudo-polynomiality result that can be used for isolating polynomial-time solvable fragments, which is possible even for optimal planning. We prove **PSPACE**-membership of the corresponding planning problem, showing that, computationally, it is no harder than the classical planning problem. In addition, we show that it is **NP**-hard even for tasks where actions have no preconditions, in contrast to classical planning, where empty preconditions imply polynomial-time solvability.

Building on Thm.1, we look into a restricted form of SNT with a propositional variable $v$ and a numeric variable $x$ such that $x$ *depends* on $v$, but not vice versa. As we can enumerate all simple paths traversing the domain of $v$, we can compute a bounded interval $I$ of $x$ for each of them, which implies a finite overall state space. We generalize this by showing that SNP with an arbitrary number of numeric leaf variables is decidable, and lies in **PSPACE** if the state space induced by the propositional variables has a fixed size. The already mentioned work by Helmert (2002) lays important foundations for our analysis. We discuss the connections in a dedicated related work section. We conclude the paper by discussing our findings and indicating directions for future research. In particular, we believe that our results can be utilized by techniques known from classical planning that require bounded domain sizes, such as abstraction heuristics based on projections onto a subset of variables (Culberson and Schaeffer 1998; Edelkamp 2001), or decoupled search, which decomposes planning tasks and requires bounded state spaces for the *leaf* components (Gnad and Hoffmann 2018).

## Numeric Planning

We consider *simple numeric planning* (SNP) as introduced by Hoffmann (2003) and refined by Scala et al. (2016). SNP

was originally defined in the STRIPS formalism (Fikes and Nilsson 1971) but we present it (in a more general way) as an extension of the finite-domain planning (FDR) formalism (Bäckström and Nebel 1995; Helmert 2009).

A *simple numeric task* (SNT) is a 4-tuple $\Pi = \langle \mathcal{V}, \mathcal{A}, s_0, G \rangle$, where $\mathcal{V} = \mathcal{V}_n \cup \mathcal{V}_p$ is a finite sets of *numeric* and *propositional variables*, $\mathcal{A}$ is the set of *actions*, $s_0$ is the *initial state*, and $G$ is the set of *goal conditions*. Numeric variables $\mathcal{V}_n$ have domain $\mathbb{Q}$; each propositional variable $v \in \mathcal{V}_p$ has a finite domain $\mathcal{D}(v)$. The set of *states* of $\Pi$ is $\mathcal{S} := \times_{v \in \mathcal{V}_p} \mathcal{D}(v) \times \times_{v \in \mathcal{V}_n} \mathbb{Q}$, i.e., a state is a full assignment over all variables $\mathcal{V}$. We refer to a state $s \in \mathcal{S}$ as a set of numeric and propositional facts $s = s_p \cup s_n$, where $s_p = \{\forall v \in \mathcal{V}_p \exists! d \in \mathcal{D}(v) : \langle v, d \rangle\}$, and similarly $s_n = \{\forall v \in \mathcal{V}_n \exists! q \in \mathbb{Q} : \langle v, q \rangle\}$. We say that $s \models (v = f)$ iff $\langle v, f \rangle \in s$, and write $s[v] = f$, i.e., $s[v]$ indicates the value of $v \in \mathcal{V}$ in state $s$. We say that $s'$ is a *partial state* if there is a state $s \in \mathcal{S}$ such that $s' \subset s$.

Conditions can be either propositional or numeric. Propositional conditions are partial propositional states, i.e., $\psi$ is a *propositional condition* if there is $s \in \mathcal{S}$ such that $\psi \subseteq s_p$. A *linear numeric condition* over the numeric variables $V \subseteq \mathcal{V}_n$ is written as $\psi : \sum_{v \in V} w_v v \unrhd w_0$ where $\unrhd \in \{\geq, >\}$, $w_v, w_0 \in \mathbb{Q}$. It is *satisfied* by $s$, denoted $s \models \psi$, if $\sum_{v \in V} w_v s[v] \unrhd w_0$. We extend this to sets of conditions $\Psi$ by $s \models \Psi$.

An *action* $a \in \mathcal{A}$ is a tuple $\langle \mathsf{pre}(a), \mathsf{eff}(a) \rangle$, where $\mathsf{pre}(a)$ are the *preconditions*, and $\mathsf{eff}(a)$ the *effects* of $a$. Preconditions are defined as $\mathsf{pre}(a) := \mathsf{pre}_p(a) \cup \mathsf{pre}_n(a)$, with propositional and linear numeric conditions, respectively. Effects $\mathsf{eff}(a) := \mathsf{eff}_p(a) \cup \mathsf{eff}_n(a)$ are similarly defined as sets of propositional and numeric effects. For SNT, numeric effects have the form $(v += c)$, where $v \in \mathcal{V}_n$ and $c \in \mathbb{Q} \setminus \{0\}$. Actions have at most one effect on each numeric variable. We say that action $a$ is *applicable* in state $s$ if $s \models \mathsf{pre}(a)$. The result of applying $a$ in $s$ is denoted by $s[\![a]\!] := s'_p \cup s'_n$, with $s'_p[v] = d$ if $\langle v, d \rangle \in \mathsf{eff}_p(a)$, $s'_n[v] = s_n[v] + c$ if $(v += c) \in \mathsf{eff}_n(a)$, and $s'_n[v] = s_n[v]$ otherwise.

The goal condition $G = G_p \cup G_n$ denotes propositional and numeric conditions, respectively. We say that $s_*$ is a *goal state* if $s_* \models G$. An *$s$-plan* is an action sequence $\pi$ that can be applied successively in $s$ and results in a goal state $s_* \models G$. A plan for $\Pi$ is an $s_0$-plan.

The set of all numeric conditions appearing in $\Pi$ is denoted by $\Psi(\Pi)$; by $\Psi(v, \Pi)$ we denote all numeric conditions where the variable $v \in \mathcal{V}_n$ appears. If $\Pi$ is obvious from the context, we simply write $\Psi(v)$. $\|\Pi\|$ is the number of bits needed for representing $\Pi$.

A *restricted task* (RT) is a variant of SNT where all numeric conditions are of the form: $\psi : v \bowtie w_0$, with $w_0 \in \mathbb{Q}$, $v \in \mathcal{V}_n$, and $\bowtie \in \{>, \geq, <, \leq\}$. Similar to SNT, actions can only increase or decrease variables by constant quantities (Hoffmann 2003; Scala, Haslum, and Thiébaux 2016). An SNT can be reduced to an RT with a simple translation.

**Translating SNT to RT.** Given a SNT $\Pi = \langle \mathcal{V}, \mathcal{A}, s_0, G \rangle$, we define a transformed task $\Pi^{\mathrm{RT}} = \langle \mathcal{V}^{\mathrm{RT}}, \mathcal{A}^{\mathrm{RT}}, s_0^{\mathrm{RT}}, G^{\mathrm{RT}} \rangle$ constructed as follows. For every numeric condition $\psi : \sum_{v \in V} w_v v \unrhd w_0$ in $\Psi(\Pi)$, we introduce a new numeric vari-

able $v^\psi \in \mathcal{V}_n^{\mathrm{RT}}$, with $s_0[v^\psi] = \sum_{v \in V} w_v s_0[v]$. Each $\psi$ is then replaced by $v^\psi \bowtie w_0$ and, for every action $a$ with an effect $v += c_v^a$ on a $v \in V$, a numeric effect on $v^\psi$ must be added, with the form $v^\psi += \sum_{v \in V} c_v^a$. This translation is polynomial in the number of numeric conditions. In what follows, unless stated otherwise, we assume all tasks to be in RT form.

## Integer Restricted Tasks

In this section we present integer RT tasks, a normal form which simplifies the forthcoming proofs. The transformation is very similar to the "domain simplification" of Helmert (2002), but additionally normalises initial state values to $0$ and conditions to be integer for all numeric variables. We redefine the transformation here in our terms, since we heavily rely on it for our results. We also prove the correspondence to the original task formally, which was not done in detail by Helmert (2002).

Suppose we have an RT $\Pi = \langle \mathcal{V}, \mathcal{A}, s_0, G \rangle$. In this task, any condition can be seen as a check whether $x \in \mathcal{V}_n$ belongs to a given rational interval (which is not necessarily bounded). We let $[\![l_-, l_+]\!]$ denote any closed, open, or half-open interval, where $l_- \in \{-\infty\} \cup \mathbb{Q}$, $l_+ \in \mathbb{Q} \cup \{+\infty\}$, and $l_- \leq l_+$. The precondition of each action $a$ has the form $\mathsf{pre}(a) = \{x \in [\![l_-, l_+]\!] \mid x \in \mathcal{V}_n\}$. Note that (1) $x \in [\![l_-, l_+]\!]$ is a semantic notation meaning $s \models x \in [\![l_-, l_+]\!]$ iff $s[x] \in [\![l_-, l_+]\!]$, and (2) we define conditions on *all* numeric variables replacing empty conditions with $x \in (-\infty, \infty)$. The numeric goal conditions $G_n$ have the same form. By $\mathcal{W}(x)$ we denote the set of numbers that appear in the numeric conditions on $x$, $\mathcal{W}(x) := \{l_-, l_+ \mid (x \in [\![l_-, l_+]\!]) \in \Psi(x)\}$. Note that $\mathcal{W}(x)$ is a finite set of rationals. Each action $a$ has numeric effects of the form $x += c$ with $c \neq 0$. We let $\mathcal{C}(x) := \{c \mid x += c \in \mathsf{eff}_n(a), a \in \mathcal{A}\} \subseteq \mathbb{Q}$ denote the set of additive constants affecting $x$.

We say that an RT $\bar{\Pi}$ is *integer* if for each $x \in \mathcal{V}_n$ the conditions P1–P3 hold.
P1. $\mathcal{C}(x) \cup \mathcal{W}(x) \subseteq \mathbb{Z}$,
P2. $s_0[x] = 0$, and
P3. if $(x \in [\![l_-, l_+]\!]) \in G_n$, then $[\![l_-, l_+]\!] \cap \mathbb{Z} \subseteq \mathbb{N}$.

We will show that every restricted task $\Pi$ has a corresponding integer instance $\bar{\Pi}$ that is solvable iff $\Pi$ is. Furthermore, $\bar{\Pi}$ can be computed in polynomial time. We start by showing that restricted numeric tasks are well-behaved under linear transformations. Assume $C \in \mathbb{Q} \setminus \{0\}$ and $B \in \mathbb{Q}$. We define the map $(C \cdot x + B)(\Pi)$ on $\Pi$ and an $x \in \mathcal{V}_n$ as follows: each condition of the form $x \in [\![a, b]\!]$ in $\Pi$ is replaced with $x \in [\![C \cdot a + B, C \cdot b + B]\!]$, and the effect $x += c$ is replaced with $x += C \cdot c$. The initial state $s_0[x] = x_0$ is replaced with $s_0[x] = C \cdot x_0 + B$.

**Lemma 1.** *Any plan for $\Pi$ is also a plan for $(C \cdot x + B)(\Pi)$, where $C \in \mathbb{Q} \setminus \{0\}$ and $B \in \mathbb{Q}$, and vice-versa.*

*Proof.* Let $\pi = \langle a_1, \ldots, a_n \rangle$ be a plan for $\Pi$. By $s_k$ we denote the state that corresponds to the subsequent application of the first $k$ actions of $\pi$, starting at the state $s_0$. Suppose that $x += c_k$ is the effect of action $a_k$ on $x$. Here we allow $c_k = 0$, so that each action has an effect on $x$. Thus,

$s_k[x] = s_0[x] + \sum_{i=1}^{k} c_i$. Since each $a_k$ is applicable in $s_{k-1}$, it holds that $s_{k-1}[x] = s_0[x] + \sum_{i=1}^{k-1} c_i \in [\![a_k, b_k]\!]$, where $(x \in [\![a_k, b_k]\!]) \in \mathsf{pre}(a_k)$, and $s_n[x] \in [\![a_g, b_g]\!]$, where $(x \in [\![a_g, b_g]\!]) \in G_n$. We need to show that $\pi$ is also a plan for $(C \cdot x + B)(\Pi)$. The proof is by induction, suppose that the actions were applied up to some $k - 1$. Let $s'_k$ be the resulting state, then

$$s'_{k-1}[x] = C \cdot s_0[x] + B + C \cdot \sum_{i=1}^{k-1} c_i =$$

$$C(s_0[x] + \sum_{i=1}^{k-1} c_i) + B \in [\![C \cdot a_k + B, C \cdot b_k + B]\!] \iff$$

$$s_{k-1}[x] = s_0[x] + \sum_{i=1}^{k-1} c_i \in [\![a_k, b_k]\!].$$

The claim $s'_n \models G$ is proved exactly in the same fashion. Note that all other conditions and affects were left unchanged, and the map affects only the variable $x$, thus $\pi$ is also a plan for $(C \cdot x + B)(\Pi)$.

To prove that any plan for $(C \cdot x + B)(\Pi)$ is a plan for $\Pi$, all we need is to recall that if $C \neq 0$, then the linear transformation is invertible, i.e., $(C \cdot x + B)(\Pi)$ is transformed into $\Pi$ using the linear transformation $x \mapsto \frac{1}{C}x - \frac{B}{C}$. We finish by applying the previous claim to this linear function. $\quad\square$

Let $\mathrm{LCD}(X)$ denote the *least common denominator* of a finite set $X$ of rational numbers, i.e., $\mathrm{LCD}(X)$ is the smallest number such that $\mathrm{LCD}(X) \cdot x$ is an integer for every $x \in X$.

**Corollary 1.** *For each* RT *$\Pi$, there exists an integer* RT *$\bar{\Pi}$ such that $\bar{\Pi}$ is solvable if and only if $\Pi$ is solvable.*

*Proof.* Consider the instance $\Pi_x = (C \cdot x + B)(\Pi)$. If $C = \mathrm{LCD}(\mathcal{C}(x) \cup \mathcal{W}(x))$, then P1 holds for $x$ in $\Pi_x$, and choosing $B = -C \cdot s_0[x]$ guarantees P2. We need to see to that the goal condition is positive. Let $x \in [\![l_-, l_+]\!]$ be the goal condition of $\Pi_x$. If $s_0[x] \in [\![l_-, l_+]\!]$ the task is trivial, thus replace the goal condition with $x = s_0[x]$. Otherwise, either $s_0[x] \leq l_-$ or $s_0[x] \geq l_+$. In the second case we multiply $C$ and $B$ by $-1$, and we do nothing in the first case. We finalize that claim by repeating this process for each $x \in \mathcal{V}_n$. This is can be done, since each map affects $x$ independently of all other numeric variables. $\quad\square$

Using Col. 1 we estimate the size of the integer RT we got. Finding the least common multiple of a finite set $N \subseteq \mathbb{N}$ can be done in $\log_2 \max N$ steps, i.e., it is linear in the number of bits representing $N$. Thus, we can compute the least common denominator of a set of rational numbers in linear time which leads to the following result.

**Lemma 2.** *Let $\Pi$ be an* RT. *Then, an integer* RT *$\bar{\Pi}$ can be computed in polynomial time, $\|\bar{\Pi}\| \in O(n^2)$, and the size of each number in $\bar{\Pi}$ is at most $n$ bits.*

*Proof.* If $\|\Pi\| = n$, then no number in $\Pi$ uses more than $n$ bits. The task $\bar{\Pi}$ can be computed in polynomial time since, by Cor. 1, the needed arithmetic operations can be performed in polynomial time. If two $m$-bit numbers are multiplied,

then the result can be written down using at most $2m$ bits so each number in $\bar{\Pi}$ has at most $n$ bits since the sum of their bits does not exceed $n$. Moreover, $\bar{\Pi}$ may contain at most $n$ numbers, since each number needs at least one bit so $\|\bar{\Pi}\| = O(n^2)$. $\quad\square$

In addition, we would like to show that in some specific tasks we can bound the numeric goal conditions to a closed interval. This will be helpful later on.

**Lemma 3.** *Let $\Pi$ be an integer* RT, *where $(x \geq g) \in G_n$, and each action that affects the numeric variable $x \in \mathcal{V}_n$ affects no other variables. Then, there is a $g' \geq g$ such that each plan that solves $\Pi$ contains a sub-sequence of actions that solves $\Pi'$.*

*Proof.* Let $C_+ = \max_{c \in \mathcal{C}(x)} c$. Let $\pi$ be a plan for $\Pi$, and let $s_* \models G$ be the state where $\pi$ terminates. Let $g' = g + C_+$. If $s_*[x] \in [g, g']$ we are done, otherwise, assume $s_*[x] \geq g'$. Since we start at $s_0[x] = 0$ and each action adds to $x$ at most $C_+$ there is at least one state alongside the execution of $\pi$ that lies inside $[g, g + C_+]$. Thus, $\pi$ has a prefix that achieves $x \in [g, g']$. Since all actions that affect $x$ do not affect any other variable, we can ignore all actions that affect $x$ after the first one that achieves the goal condition on $x$. $\quad\square$

If the conditions of Lemma 3 hold for $x \in \mathcal{V}_n$, we replace the goal condition $x \geq g$ with $x \in [g, g + C_+]$. In what follows, we consider integer RTs with bounded numeric goal intervals. We start with the most basic case of a single numeric variable.

## Single Numeric Variable

Suppose we have a planning task with a Single Numeric Variable (SVNT), $\Pi_x = \langle \mathcal{V}, \mathcal{A}, s_0, G \rangle$, where $\mathcal{V} = \mathcal{V}_n = \{x\}$. In this section we analyse the computational complexity of the plan existence problem for SVNTs. We denote this problem by PESVNT. We first prove that PESVNT always can be solved by search in a finite subset of the state space; this is not clear a priori from the problem formulation. This is the basis for our forthcoming complexity results.

### Finite Search Space

Since a SVNT has exactly one numeric variable, it is an RT by definition. By Cor. 1 and Lem. 3 we have that $\Pi_x$ can be transformed into an integer RT with a bounded goal condition, and by Lem. 2 it can be done in polynomial time.

We aim to show that every solvable task $\Pi_x$ has the following property: one can compute a bounded interval $I$ from $\Pi_x$ and if there exists a plan $\pi$ for $\Pi_x$, then there exists a re-ordering of $\pi$ such that $x$ stays inside $I$ during its execution. In other words, one only needs to consider a finite part of the infinite state space, hence PESVNT is a decidable problem.

**Lemma 4.** *Let $\mathcal{C} \subseteq \mathbb{Z}$ be a finite set of integers, and let $C^{\max} = \max_{c \in \mathcal{C}} |c|$. Let $a, b \in \mathbb{Z}$ be such that $|a - b| \leq 2C^{\max}$, and let $\{c_i\}_{i=1}^{n}$ be a sequence of numbers such that*

$$\forall i \in [n] : c_i \in \mathcal{C} \text{ and } a + \sum_{i=1}^{n} c_i = b.$$

*Then, for each $y \in \mathbb{R}$ such that $a, b \in [y, y + 2C^{\max}]$ there is a permutation $\sigma : [n] \to [n]$ such that for each $k \in [n]$ it holds that $a + \sum_{i=1}^{k} c_{\sigma(i)} \in [y, y + 2C^{\max}]$.*

*Proof.* The proof is by induction. Suppose that for $k - 1$ it holds that $a_{k-1} := a + \sum_{i=1}^{k-1} c_{\sigma(i)} \in [y, y + 2C^{\max}]$. For the element of the index $\sigma(k)$ we need to chose one of the indices from the set $[n] \setminus \sigma([k-1])$. We have three cases:

*Case 1.* There is $i \in [n] \setminus \sigma([k-1])$ such that $c_i = 0$. Then, set $\sigma(k) := i$, and $a_{k-1} = a_k$.

*Case 2.* All elements with indices in $[n] \setminus \sigma([k-1])$ are of the same sign. In this case, for each $i \in [n] \setminus \sigma([k-1])$ the sum $a_{k-1} + c_i$ lies on the interval between $a_{k-1}$ and $b$, which is contained in $[y, y + 2C^{\max}]$, since $a_{k-1}$ belongs to this interval by induction, and $b$ by definition. Thus, we can set $\sigma(k) := i$ for any $i \in [n] \setminus \sigma([k-1])$.

*Case 3.* There are two indices $i, j \in [n] \setminus \sigma([k-1])$ such that $c_i$ and $c_j$ are of different sign. By definition, it holds that $|c_i|, |c_j| \leq C^{\max}$. Note that since $a_{k-1} \in [y, y + 2C^{\max}]$, it either holds that $a_{k-1} - y \leq C^{\max}$ or that $y + 2C^{\max} - a_{k-1} \leq C^{\max}$, i.e., the distance from $a_{k-1}$ to one of the endpoints of the interval exceeds $C^{\max}$. Thus, either $a_{k-1} + c_i$ or $a_{k-1} + c_j$ lies in the interval $[y, y + 2C^{\max}]$. Without loss of generality, assume that $a_{k-1} + c_j \in [y, y + 2C^{\max}]$. Then, we set $\sigma(k) := j$, and repeat the process. $\square$

Given an SVNT $\Pi_x$ and a bounded interval $I$, we let $\Pi_x^I$ denote $\Pi_x$ where each precondition $\mathsf{pre}(a) = \{x \in [\![l_-, l_+]\!]\}$ of each action $a \in \mathcal{A}$ is replaced with the precondition $\mathsf{pre}(a) = \{x \in [\![l_-, l_+]\!] \cap I\}$.

**Theorem 1.** *Let $\Pi_x$ be an SVNT. Then there exists a bounded interval $I = [M_- - 2C^{\max}, M_+ + 2C^{\max}]$ such that each plan $\pi$ for $\Pi_x$ can be reordered into a plan $\pi'$ for $\Pi_x^I$. The constants are $C^{\max} = \max_{c \in \mathcal{C}(x)} |c|$, $M_- = \min(\mathcal{W}(x) \cup \{0\}) - 1$, and $M_+ = \max(\mathcal{W}(x) \cup \{0\}) + 1$.*

*Proof.* Let $\pi$ be a plan for $\Pi_x$, and $\langle s_0, \ldots, s_n \rangle$ be the sequence of states traversed by $\pi$. Intuitively, we would like to reorder not the whole plan $\pi$, but only the parts of the plan that exceed $M_+$ from above and $M_-$ from below. Let us look at the following state\action sequence

$$s_0, a_1, s_1, a_2, \ldots s_{k_1}, a_{k_1+1}, s_{k_1+1}, \ldots, a_{k_2}, s_{k_2+1}, \ldots,$$
$$a_{k_3}, \ldots, a_{k_4}, \ldots, a_{k_5}, \ldots, a_{k_6}, \ldots, s_n \models G.$$

We say that an action exceeds $M_+$ if both the start and the end state of this action exceed $M_+$ from above. The definition for exceeding $M_-$ is the same, but the states should exceed $M_-$ from below. Suppose that the red sub-sequences in the plan are the ones that exceed $M_+$ and the blue ones exceed $M_-$. Note that, since $M_- < M_+$, between each red and blue sub-sequence, there must be at least one black action. The first and the last actions in the sequence are also black, since, by definition, both $s_0[x] = 0$ and the interval $[g_1, g_2]$ lie inside the interval $[M_-, M_+]$, where $G = \{x \in [g_1, g_2]\}$.

Let $I = [L_-, L_+]$. First, we obtain the $L_-$ bound. Let $\mathcal{A}^{-\infty} := \{a \in \mathcal{A} \mid \forall s[x] \leq M_- : s \models \mathsf{pre}(a)\}$. So $\mathcal{A}^{-\infty}$ is the set of all actions that have as precondition either $x \in \mathbb{Q}$ (no preconditions) or $x \leq b$ for some $b \in \mathbb{Q}$, i.e., the actions that can be applied with an arbitrarily small $x$.

If no state in $\langle s_0, \ldots, s_n \rangle$ achieves a value below $M_-$, there is no need for reordering. Otherwise, let $s_{k_1}$ be the first state below $M_-$, and let $\pi_{k_1 \to k_2} := \{a_{k_1+i}\}_{i=1}^{k_2-k_1}$ be the longest sub-sequence of $\pi$ that starts in $s_{k_1}$ such that all actions in $\pi_{k_1 \to k_2}$ exceed $M_-$. Let $s_{k_2}$ be the last state along the application of $\pi_{k_1 \to k_2}$. Note that it may happen that $s_{k_1} = s_{k_2}$, in this case $\pi_{k_1 \to k_2} = \emptyset$. By definition, $s_{k_1} \neq s_0$ and $s_{k_2} \not\models G$. Since each action cannot increase or decrease the value of $x$ by more than $C^{\max}$, it holds that $s_{k_1}[x], s_{k_2}[x] \in [M_- - C^{\max}, M_-]$. Thus, $|s_{k_1}[x] - s_{k_2}[x]| \leq C^{\max}$. Moreover, since $\pi_{k_1 \to k_2}$ is a sequence of action applications:

$$s_{k_1}[x] + \sum_{i=1}^{k_2-k_1} c_{k_1+i} = s_{k_2}[x],$$

where $c_j$ is the effect of applying the action $a_j$ in the state $s_{j-1}$. By construction, all actions in $\pi_{k_1 \to k_2}$ belong to $\mathcal{A}^{-\infty}$, and thus can be applied within the interval $(-\infty, M_-]$. By Lem. 4, there exists a permutation $\sigma$ of indices of $\pi_{k_1 \to k_2}$ such that

$$s_{k_1}[x] + \sum_{i=1}^{k_2-k_1} c_{k_1+\sigma(i)} \in [M_- - 2C^{\max}, M_-].$$

Set $L_- := M_- - 2C^{\max}$. We remark that there is a finite number of such disjoint prefixes, and each prefix can be reordered in such a way that for each state $s$ along the reordered plan, it holds that $s[x] \geq L_-$.

The reordering that bounds the value of $x$ along $\pi$ from above is obtained in the same way, taking the upper bound $L_+ := M_+ + 2C^{\max}$ instead. $\square$

## Complexity Results

Armed with Thm. 1, we are ready to prove concrete complexity results. We begin with a pseudopolynomiality result.

**Theorem 2.** *The problem* PESVNT *can be solved in pseudopolynomial-time* $O(|\mathcal{A}|(C^{\max} + W^{\max}))$, *where* $C^{\max}$ *and* $W^{\max}$ *are the maximums over the absolute values of the sets* $\mathcal{C}(x)$ *and* $\mathcal{W}(x)$, *respectively.*

*Proof.* Let $\Pi_x$ be an instance of PESVNT. Let $\bar{\Pi}_x$ be the normalised version of $\Pi_x$ via Lem. 2. By Thm. 1, we can construct a bounded-interval SVNT $\bar{\Pi}_x^I$ such that for each plan for $\bar{\Pi}_x$ there is a plan for $\bar{\Pi}_x^I$. By construction of the interval $I$ we have that $|I \cap \mathbb{Z}| \leq 4C^{\max} + 2W^{\max}$. Note that $x$ in the task $\bar{\Pi}_x^I$ can take only integer values since $s_0[x] = 0$ and all additive constants are integers. Thus, to find a plan for $\bar{\Pi}_x^I$ we can use dynamic programming, breaking down the problem into sub-problems of finding the shortest path from $x = 0$ to any of the integers in the interval $I$. Hence, the complexity is $O(|\mathcal{A}|(C^{\max} + W^{\max}))$. $\square$

If $X$ is a set of SVNTs with bounded $C^{\max}$ and $W^{\max}$, then the corresponding planning problem can be solved in polynomial time by Thm. 2. We additionally note that the dynamic programming approach can be used for finding an optimal plan; recall that the plan obtained by Thm. 1 has at most the length (and cost) as the original plan. We continue

by proving that if the parameters are not bounded, then the problem becomes **NP**-hard. We also present a positive result: PESVNT is in **PSPACE** so it is not computationally harder than the FDR planning problem.

To prove membership in **PSPACE** we need the following famous result by Savitch (1970). We remind that **NSPACE**$(f(n))$ is the class of all decision problems that can be solved by nondeterministic algorithms using space $O(f(n))$, while **DSPACE**$(f(n))$ is defined the same but for deterministic algorithms.

**Theorem 3.** *Suppose that $f(n)$ can be computed in $O(f(n))$ time. Then, **NSPACE**$(f(n)) \subseteq$ **DSPACE**$(f(n)^2)$.*

We use this theorem to prove the following result.

**Theorem 4.** *The problem PESVNT is in **PSPACE**, and it is **NP**-hard even if all preconditions are empty.*

*Proof. Membership* in **PSPACE**: Let $\Pi_x$ be an SVNT with $\|\Pi_x\| = n$. By Lem. 2, $\|\bar{\Pi}_x\| \in O(n^2)$ and $C^{\max}, W^{\max}$ use at most $2n$ bits. By Thm. 1 there is an interval $I = [-W^{\max} - 2C^{\max}, W^{\max} + 2C^{\max}]$ such that $\bar{\Pi}_x$ is solvable iff $\bar{\Pi}_x^I$ is.

By the proof of Thm. 2, the size of the search space of $\bar{\Pi}_x^I$ is bounded by the size of the interval $I$, i.e., $4C^{\max} + 2W^{\max}$, which is, in turn, at most $4 \cdot 2^{2n} + 2 \cdot 2^{2n} \leq 2^{2n+3}$. Thus, any solution of length $2^{2n+3}$ or larger has cycles. Such cycles can be removed, resulting in a solution of length less than $2^{2n+3}$. We can guess a plan one action at a time and verify it step by step using only polynomial space. No more than $2^{2n+3}$ non-deterministic choices are required so this non-deterministic algorithm uses only polynomial space. By Savitch's theorem (1970), it holds that **NPSPACE** = **PSPACE** so PESVNT is in **PSPACE**.

**NP**-*hardness*. The basis for our proof is the feasibility version of the *change-making problem* (FEAS-CMP) (Martello and Toth 1990, Sec. 5). An instance is an $n$-vector $(c_1, \ldots, c_n)$ of non-negative integers and a non-negative integer $b$. The question is if there are non-negative integers $x_1, \ldots, x_n$ such that $\sum_{i=1}^n c_i \cdot x_i = b$? The **NP**-hardness of this problem was originally proved by Lueker (1975).

We present a polynomial-time reduction from FEAS-CMP to PESVNT. Let $(\bar{c}, b)$ denote an arbitrary instance of this problem with $\bar{c} = (c_1, \ldots, c_n)$. We construct a planning task $\Pi_x$ as follows: introduce a variable $x$ and actions $a_i$, $i \in [n]$, with empty preconditions and effects $x \mathrel{+}= c_i$. The initial state is $x = 0$ and the goal state is $x = b$. The task $\Pi_x$ can be constructed in polynomial time and $(\bar{c}, b)$ has a solution iff $\Pi_x$ is solvable. $\qquad\square$

We note that FDR planning is strongly NP-hard even in severely restricted cases (see Figure 3 in (Bäckström and Nebel 1995)). Thus, pseudopolynomial algorithms for FDR are ruled out under standard assumptions. This is a quite noticeable difference between FDR planning and PESVNT (that are closely related by both being in **PSPACE**).

## Numeric Causal-Graph Leaf Variables

Planning tasks are typically structurally complex, and *causal graphs* are a common means to study this structure (e.g.,

Knoblock 1994; Bacchus and Yang 1994; Domshlak and Brafman 2002). In this paper we use the compact definition by Helmert (2004): the *causal graph* (CG) of a classical planning task $\Pi = \langle \mathcal{V}, \mathcal{A}, I, G \rangle$ is a digraph $CG(\Pi) = \langle \mathcal{V}, \mathcal{E} \rangle$, where $(u, v) \in \mathcal{E}$ if $u \neq v$ and there exists $a \in \mathcal{A}$, s.t. $u \in vars(\mathsf{pre}(a)) \cup vars(\mathsf{eff}(a))$ and $v \in vars(\mathsf{eff}(a))$, where $vars(s)$ denotes the set of variables defined in the (partial) state $s$. Intuitively, the CG contains an edge from a variable $v$ to a variable $v'$, if changing the value of $v'$ might require $v$ to change its value, too, so $v'$ *depends* on $v$.

For general numeric planning, it is not immediately clear how to adopt this definition. We start with the more obvious case of an RT. As every precondition and effect of an action $a \in \mathcal{A}$ of an RT touches at most one numeric variable, the influence of $a$ on the CG of the task is the same as one of an action with propositional effects. Thus, we can treat numeric variables exactly as propositional ones for CG construction.

The case of linear conditions $\psi : \sum_{x \in V} w_x s[x] \unrhd w_0 \in \mathsf{pre}_n(a)$ in SNT is somewhat more intricate. If numeric variables $x_1$ and $x_2$ both appear in $V$, they are co-dependent in terms of $\psi$. Note that the case $x_1 + x_2 \geq w_0$ differs from the case when $x_1 \geq w_1$ and $x_2 \geq w_2$. This co-dependency is also shown by the fact that the translation to RT, which introduces a new variable $v^\psi$, introduces cycles in the causal graph between $x_1, x_2$, and $v^\psi$. How to define a CG in this cases remains an open question.

In this work we consider only the case of RT. Moreover, we concentrate on numeric variables that are *leaves* in the causal graph, i.e., they might depend on other variables via action preconditions, but no other variable depends on their value and there are no co-effects. We call numeric variables that are leaves in the causal graph *numeric leaves*. In what follows, all numeric variables are numeric leaves. We further say that a CG has a *fork structure*, or simply *is a fork*, if there exists a variable $v$ such that all edges in the causal graph are of the form $v \to x$, and there is an edge for each $x \in \mathcal{V}_n$. The notion of forks was introduce in classical planning by Katz and Domshlak (2008). To our best knowledge this work is the first work that uses causal graphs in the setting of numeric planning.

## Forks have a Finite Search Space

We start with a claim that constitutes a basic building block to the complexity results below, when there exists only a single numeric leaf variable that depends on a single propositional variable.

**Lemma 5.** *Let $\Pi_{v,x}$ be an RT with two variables $\mathcal{V} = \{v, x\}$, where $v \in \mathcal{V}_p$ is a propositional variable, and $x \in \mathcal{V}_n$ is a numeric variable. Assume further that the causal graph of the task has exactly one edge $(v, x)$. Then, the plan existence problem for $\Pi_{v,x}$ is decidable.*

*Proof.* We assume that $\Pi_{v,x}$ is an integer task with bounded numeric goal condition, since by Cor. 1, Lem. 2, and Lem. 3 $\Pi_{v,x}$ can be transformed to this form in quadratic time.

We start with the observation that each action $a$ can not affect both $v$ and $x$ due to the CG structure of the task. Hence, if an action affects the propositional variable $v$ it has the form $\mathsf{pre}(a) = \{\langle v, u \rangle\}$ and $\mathsf{eff}(a) = \{\langle v, u' \rangle\}$, where

$u, u' \in \mathcal{D}(v)$. Thus, all actions affecting $v$ are *inner actions*. We can view the values of $v$ as a directed graph where the nodes are values of $v$, and each action with precondition $\mathsf{pre}(a) = \{\langle v, u\rangle\}$ and effect $\mathsf{eff}(a) = \{\langle v, u'\rangle\}$ corresponds to a directed edge $(u, u')$. We denote the $k \in \mathbb{N}$ strongly connected components (SCC) of this graph by $\{\mathcal{C}_j\}_{j=1}^k$. Recall, that the SCCs of a graph induce a DAG. For each fact $\langle v, u\rangle$ that belongs to an SCC $\mathcal{C}_j$ we write $u \in \mathcal{C}_j$.

We denote the constants $C_x^{\max} = \max_{c \in \mathcal{C}(x)} |c|$, $M_- = \min(\mathcal{W}(x) \cup \{0\}) - 1$, and $M_+ = \max(\mathcal{W}(x) \cup \{0\}) + 1$.

Our goal for the proof is to compute an interval $[L_-^x, L_+^x]$ s.t. for each plan $\pi$ there exist a plan $\pi'$ where for all states $s$ traversed by $\pi'$ it holds that $s[x] \in [L_-^x, L_+^x]$. Intuitively, we would like to reconstruct not the whole, but only the parts of the plan, where the value of $x$ exceeds $M_+$ from above or $M_-$ from below. Let $\pi$ be a plan for $\Pi_{v,x}$, and $\langle s_0, \ldots, s_n\rangle$ be the sequence of states traversed by $\pi$.

Consider the following state/action sequence:

$$s_0, a_1, s_1, a_2, \ldots s_{k_1}, a_{k_1+1}, s_{k_1+1}, \ldots, a_{k_2}, s_{k_2+1}, \ldots,$$
$$a_{k_3}, \ldots, a_{k_4}, \ldots, a_{k_5}, \ldots, a_{k_6}, \ldots, s_n \models G.$$

We say that an action *exceeds* $M_+$ if both in the start and the end state of this action the value of $x$ exceeds $M_+$ from above. The definition for exceeding $M_-$ is the same, but the states should exceed $M_-$ from below.

If no state in $\langle s_0, \ldots, s_n\rangle$ achieves a value below $M_-$, there is no need for reconstruction. Otherwise, let $s_{k_1}$ be the first state below $M_-$, and let $\pi_{k_1 \to k_2} := \{a_{k_1+i}\}_{i=1}^{k_2-k_1}$ be the longest sub-sequence of $\pi$ that starts in $s_{k_1}$ such that all actions in $\pi_{k_1 \to k_2}$ exceed $M_-$. Let $s_{k_2}$ be the last state along the application of $\pi_{k_1 \to k_2}$. Note that it may happen that $s_{k_1} = s_{k_2}$, in this case we set $\pi_{k_1 \to k_2} = \emptyset$. Otherwise, by definition, $s_{k_1} \neq s_0$ and $s_{k_2} \not\models G$. By construction of the sub-sequence it holds that $s_{k_1-1}[x], s_{k_2+1}[x] < M_-$, thus $s_{k_1}[x], s_{k_2}[x] \in [M_- - C_x^{\max}, M_-]$. Define $a := s_{k_1}[x]$ and $b := s_{k_2}[x]$. The actions in $\pi_{k_1 \to k_2}$ traverse the values of $v$ in some order over its SCCs, assume this order to be

$$\vec{\mathcal{C}} : \mathcal{C}_1 \to \mathcal{C}_2 \to \cdots \to \mathcal{C}_m.$$

We next show how to compute a bound $\mathcal{M}_-^{\vec{\mathcal{C}},a,b}$ for all possible value pairs $a, b \in [M_- - C_x^{\max}, M_-] \cap \mathbb{N}$ and all possible SCC-chains $\vec{\mathcal{C}}$. Thus, for each sub-sequence $\pi_{k_1 \to k_2}$ of the plan $\pi$ that exceeds $M_-$ from below there will be a bound

$$L_-^x = \min_{\vec{\mathcal{C}};a,b \in [M_- - C_x^{\max}, M_-]} \mathcal{M}_-^{\vec{\mathcal{C}},a,b}$$

that bounds a reconstructed plan from below. We can compute $L_-^x$ by iteratively solving an appropriate minimization problem, which we explain next.

We need to check if there exists an action sequence that drives $x$ from $a$ to $b$ without exceeding $M_-$. We iterate over all possible sequences of SCCs. Let $\vec{\mathcal{C}}$ be such a sequence. Define for each $i \in [m]$ the following sets of actions:

$$\mathcal{A}_i^x = \mathcal{A}_0^x \cup \{a \in \mathcal{A} \mid \mathsf{pre}(a) = \{\langle v, u\rangle\}, x \in (-\infty, w_a]\},$$
$$M_- \leq w_a, u \in \mathcal{C}_i\}, \text{ where}$$
$$\mathcal{A}_0^x = \{a \in \mathcal{A} \mid \mathsf{pre}(a) = \{x \in (-\infty, w_a]\}, M_- \leq w_a\}.$$

Note that all actions in $\mathcal{A}_i^x$ can be applied interchangeably. Define the following minimization problem on the number of times $n_a^j$ an action $a$ with additive effect $c_a$ is applied while $v$ has a value in $\mathcal{C}_j$ as follows:

$$\max f(n) := \sum_{j=1}^m \sum_{a \in \mathcal{A}_j^x : c_a < 0} n_a^j c_a,$$

$$\text{s.t.} \sum_{j=1}^m \sum_{a \in \mathcal{A}_j^x} n_a^j c_a = b - a, \qquad (\heartsuit)$$

$$\sum_{j=1}^k \sum_{a \in \mathcal{A}_j^x} n_a^j c_a \leq M_- \qquad (\forall k \in [m-1]),$$

$$n_a^j \in \mathbb{N} \qquad (\forall a \in \mathcal{A}, j \in [m]).$$

It is important to note that we can apply the actions in $\mathcal{A}_i^x$ in any order we want (modulo the applications of inner actions of $v$). Thus, by Lemma 4 if the minimization problem has a solution, there is a sequence of actions that leads from $a := s_{k_1}[x]$ to $b := s_{k_2}[x]$, where for each state along the application of this sequence the value of $x$ does not exceed $M_-$. Note that we want to maximize $f(n)$ since all considered $c_a$ are negative, so we attempt to get as close to 0 as possible.

Let $n^*$ be the optimal solution for the optimisation problem. By construction, any ordering of action applications can reach a point that is less that $f(n^*)$, thus we set

$$\mathcal{M}_-^{\vec{\mathcal{C}},a,b} := f(n^*).$$

By solving this problem for all possible $\vec{\mathcal{C}}$, $a$, and $b$, we can obtain the lower bound $L_-^x$. The number of optimization problems we need to solve to obtain $L_-^x$ is then $2^N (C_x^{\max})^2$, where $N$ is the number of SCCs in the domain of $v$.

The solution for $L_+^x$ is almost the same but we replace $<$ with $>$ and $\max$ with $\min$ in the optimization problems. $\square$

This result brings us an important step forward into analyzing more general numeric planning tasks. In the next section, we generalize the result to tasks with multiple numeric leaves.

## PSPACE Membership of Forks

Let $\Pi_{v,x_1,\ldots,x_n}$ be an RT with a propositional variable $v$ and numeric leaf variables $x_i$. Note that we can transform $\Pi_{v,x_1,\ldots,x_n}$ into an integer RT with bounded numeric goal conditions in polynomial time. We consider such transformed tasks in the following and denote them by FRT. The plan existence problem for FRT is denoted by PEFRT.

We show that deciding if the modified task $\Pi_{v,x_1,\ldots,x_n}$ is solvable lies in **PSPACE**. To show this, we need the following result by Papadimitriou (1981, Lemma 4). We assume that the coefficients in integer linear programs (ILP) always are integers.

**Lemma 6.** *If the following ILP program is feasible and bounded, then for its optimal solution $z^*$ holds that $|z^*| \leq$*

$M \cdot \sum_{i=1}^{t} |c_i|$.

$$\min c'x$$
$$\text{s.t. } Ax \leq b, x \in \mathbb{N},$$

*Here, $M = t^2(ma^2)^{2m+3}$, where $m \times t$ is the size of the integer program, and $a = \max_{i \in [t], j \in [m]}\{|a_{i,j}|, |b_i|\}$ is a bound on the sizes of numbers in the program.*

**Theorem 5.** *PEFRT lies in **PSPACE**.*

*Proof.* Let $\Pi_{v,x_1,\ldots,x_n} = n$ be an FRT of a size $n$, implying $|\mathcal{D}(v)| \leq n$. Let further

$$\vec{\mathcal{C}} : \mathcal{C}_1 \to \mathcal{C}_2 \to \cdots \to \mathcal{C}_m$$

be a path through the SCCs in the domain of $v$. To show the claim we iterate over all possible such paths, of which there are at most $2^n$, fix one $\vec{\mathcal{C}}$ at a time, bound the problem, and check for solvability. Note that if $\Pi_{v,x_1,\ldots,x_n}$ has a solution, it must traverse a $\vec{\mathcal{C}}$. Such a plan $\pi$ needs to respect $\vec{\mathcal{C}}$, so $\langle v, u \rangle \in s_0 \cap \mathcal{C}_1$, and, if $\langle v, u' \rangle \in G$, we also require $\langle v, u' \rangle \in \mathcal{C}_m$. Note that for a given $\vec{\mathcal{C}}$ we can ignore actions with preconditions and effects in $v$ that do not obey $\vec{\mathcal{C}}$.

With a fixed $\vec{\mathcal{C}}$, we can use Lemma 5 to individually bound the domains of $\{x_i\}_{i=1}^n$. For each $x \in \{x_i\}_{i=1}^n$ the bound from below for an $\vec{\mathcal{C}}$ is given by a solution of $(C_x^{\max})^2$ ILPs, and constitutes

$$L_-^{x,\vec{\mathcal{C}}} = \min_{a,b \in [M_- - C_x^{\max}, M_-]} \mathcal{M}_-^{\vec{\mathcal{C}},a,b}$$

We have that $C_x^{\max} \leq 2^n$ since it is part of the input. Iterating over all possible $a$ and $b$ we solve the ILP optimization problem given in the proof of Lemma 5. Note that all constants except $b - a$ in the ILP come from the definition of the problem, and $|b - a| \leq C_x^{\max} \leq 2^n$. The solution of the problem is bounded from above by 0 ($c_a < 0$ and $n_a \geq 0$), thus if the program is feasible it has an optimal solution, which we denote by $z^*$. Now, we apply Lemma 6. The number of constraints in this program (see $\heartsuit$) equals the number of SCCs traversed by $\vec{\mathcal{C}}$, so is bounded by $n$; the number of variables is $|\vec{\mathcal{C}}| \cdot |\mathcal{A}|$, since some actions can appear in all connected components. We can bound the number of variables by $n^2$. Hence, by Lemma 6 we have

$$|z^*| \leq n^4 (n2^{2n})^{2n+3} \cdot \sum_{i=1}^{n} 2^n = n^{2n+8} 2^{4n^2+7n}.$$

Thus, we can bound $|z^*|$ by $2^{4n^2+8n}$ for sufficiently large $n$.

To obtain an upper bound on the values taken by $x$, we compute $L_+^{x,\vec{\mathcal{C}}}$ in the same fashion. Hence, the domain of the numeric variable $x$ under the path $\vec{\mathcal{C}}$ is of the size $2^{4n^2+8n+1}$.

The whole state space given the path $\vec{\mathcal{C}}$ can be bounded by

$$|\mathcal{D}(v)| \cdot \Pi_{i=1}^n |\mathcal{D}(x_i)| \leq n2^{\sum_{i=1}^n 4n^2+8n+1} = n2^{4n^3+8n^1+n}$$

Finally, to account for the choice of $\vec{\mathcal{C}}$ and get the whole search space we need to multiply by the number of directed paths $\vec{\mathcal{C}}$, which also cannot exceed $2^n$. Hence, the whole state space can not exceed $2^{5n^3}$ for sufficiently large $n$.

Thus, by Savitch's theorem, we can guess a solution of the task $\Pi_{v,x_1,\ldots,x_n}$ in polynomial space. □

Thm. 5 is a major step forward in our analysis, showing tasks with fork causal graph that may have an arbitrary number of numeric variables to be computationally no harder than classical planning.

**General Causal Graphs with Numeric Leaves**

In this section, we further generalize the previous result to tasks with multiple propositional variables, where all numeric variables are causal-graph leaves. Let $\Pi$ be such an RT task. Note that since all numeric variables of $\Pi$ are leaves in the CG, we can transform the RT into its integer form with bounded goal conditions for all numeric variables. We denote such tasks by NLRT and the corresponding plan existance problem by PENLRT.

**Theorem 6.** *PENLRT is decidable.*

*Proof.* Recall that by Thm. 5, checking plan existence for an RT with a with a fork-structured CG with numeric leaves is in **PSPACE**, so decidable. Here, we show that we can transform an NLRT to an FRT using exponential space. While this transformation does not preserve membership in **PSPACE**, it is sufficient to show that PENLRT is decidable.

Let us look at all partial propositional states of the task $\Pi$, $\mathcal{S}(\mathcal{V}_p) := \times_{v \in \mathcal{V}_p} \mathcal{D}(v)$. We can replace all propositional variables in $\Pi$ with a single variable $s$ s.t. $\mathcal{D}(s) = \mathcal{S}(\mathcal{V}_p)$. Each action that affects a $v \in \mathcal{V}_p$ is transformed to an inner action of $s$. The transformation may produce an exponential number of actions, since $\mathcal{S}(\mathcal{V}_p)$ is a projection of the state transition graph on $\mathcal{V}_p$, and thus may have exponential number of partial states and inner actions in the size of $|\mathcal{V}_p|$.

The actions that affect any variable $x \in \mathcal{V}_n$ do not affect any other variable $v \in \mathcal{V}$, because this was already the case in the original task. Hence we know that the transformed task has one propositional CG-root variable $s$ with a potentially exponential-size domain, and numeric variables $x_1, \ldots, x_n$ that may depend on $s$. Hence, each $x \in \mathcal{V}_n$ is either a CG-leaf, or constitutes a singleton. Since separated components of a CG can be solved separately, we invoke Thm. 5 to check for the decidability of the fork, and Thm. 1 to check for the decidability of singletons. □

Our last result shows **PSPACE**-membership if we fix the number of propositional variables. We denote by $k$-PENLRT the plan existence problem for NLRT with a fixed number $k$ of propositional variables.

**Corollary 2.** *$k$-PENLRT is in **PSPACE**.*

*Proof.* Let $\Pi$ be an NLRT where $\|\Pi\| = n$ and $|\mathcal{V}_p| = k$. Let $\mathcal{S}(\mathcal{V}_p)$ be the domain of the new CG-root variable as introduced in the proof of Thm. 6. Note that since the domains of our propositional variables are not bounded we have that $|\mathcal{S}(\mathcal{V}_p)| \in O(n^k)$. This is due to the fact that for each $v \in \mathcal{V}_p$ we have $|\mathcal{D}(v)| \leq n$. The number of actions in the new task is $O(|A| \cdot n^k) = O(n^{k+1})$, since every new action may have at most one precondition in $\mathcal{S}(\mathcal{V}_p)$. By plugging this number into the last part of the proof of Thm. 5, we have that the state space of the transformed task is $2^{O(n^{k+c})}$ for some small universal constant $c \in \mathbb{N}$. The overall state space of the bounded numeric RT is bounded by $|\mathcal{S}(\mathcal{V}_p)| \cdot \max_{i \in [n]} |\mathcal{D}(x_i)|^n$ so

we need to bound $|\mathcal{D}(x_i)|$ for $x_i \in \mathcal{V}_n$. As before, we apply Lem. 6 to the ILP $\heartsuit$ from the proof of Lem. 5. In our case, the number of constraints is equal to the number of SCCs of $\mathcal{S}(\mathcal{V}_p)$, $m := n^k$. The number of variables is the number of actions in the new task times the number of SCCs, that is $t := n^{2k+1}$. The maximal constant of the problem is $a := 2^n$. Thus, the size of the state space is bounded by

$$2t^2(ma^2)^{2m+3} = 2 \cdot n^{2k+1}(n^k 2^{2n})^{2n^k+3} =$$
$$O(2^{5n^{k+1}}) \subseteq 2^{O(n^{k+1})}.$$

The multiplication by 2 takes into account that we solve two ILPSs: one for the positive bound and one for the negative bound. Take this bound to the power of $n$ so that we cover all numeric variables and we see that the universal constant $c = 2$, since $(2^{O(n^{k+1})})^n = 2^{nO(n^{k+1})} = 2^{O(n^{k+2})}$. Thus, by Savitch's theorem (1970), $k$-PENLRT is in **PSPACE**. $\square$

The generalization to multiple propositional variables has important practical implications, since it allows to augment arbitrary classical planning tasks with many numeric leaf variables, without affecting the computational complexity.

## Related Work

We are only aware of one important piece of related work, the complexity analysis by Helmert (2002). As indicated before, that work focused on investigating the decidability boundaries of general numeric planning as defined by the "level 2" of PDDL2.1 (Fox and Long 2003). In their analysis, Helmert (2002) consider different levels of expressiveness of the mathematical formulas allowed in conditions and effects. The simple numeric planning formalism considered in our work corresponds to their class $(\mathcal{C}_c, \mathcal{C}_c, \mathcal{E}_{\pm c})$, i. e., the class of tasks where numeric variables can be compared to constants in preconditions and the goal, and effects can only add constants. This was proved to be undecidable.

In our work, we introduced a normal form for SNP tasks that is similar to the domain simplification of Helmert (2002). We outlined the differences between the two in the section on Integer Restricted Tasks. Besides this, there is a weak connection between the constructions in some of our proofs and their Algorithm 22. Both approaches rely on guessing action sequences, and both employ an ILP that encodes the number of times certain actions are applied. The details of how these approaches are used differ significantly, though. While Helmert (2002) only requires the number of guesses to be bounded to show decidability, we derive a bound that shows **PSPACE**-membership.

## Discussion

Our results are a major step forward in the complexity analysis of simple numeric planning. Prior results by Helmert (2002) show that numeric planning is undecidable even for highly restricted instances, but that work focused on the expressive power of the mathematical expressions that are used in conditions and effects. In this work, we analyzed the impact of the number of numeric variables and the causal structure of planning tasks. We investigated the case of a single numeric variable as a basis for our more advanced results,

showing that it is **NP**-hard in general. With this, we proved that SNP is decidable as long as the numeric variables are leaf nodes in the causal graph. The decision problem even lies in **PSPACE** for a fixed number of propositional variables, so it stays in the class of classical planning.

Beyond the theoretical insights, we believe that our results can have a relevant practical impact. Many techniques known from classical planning rely on finite domain sizes, so are not directly applicable to unbounded numeric variables. With our findings, we can provide exactly these bounds. Hence, techniques such pattern database heuristics (Culberson and Schaeffer 1998; Edelkamp 2001), which project onto a subset of the variables, and decoupled search (Gnad and Hoffmann 2018), which requires bounded leaf components, can probably be adopted to numeric planning.

For the future, we intend to further continue both "branches" of our analysis. The complexity of SNP tasks with two numeric variables is still unknown (the case for three or more variables is undecidable, which follows from a result of Helmert 2002). Moreover, looking into different causal structures is obviously highly interesting.

## Acknowledgements

The work of Alexander Shleyfman is partially supported by the Israel Academy of Sciences and Humanities program for Israeli postdoctoral researchers. Daniel Gnad was partially supported by the Wallenberg AI, Autonomous Systems and Software Program (WASP) funded by the Knut and Alice Wallenberg Foundation, and by TAILOR, a project funded by the EU Horizon 2020 research and innovation programme under grant agreement no. 952215. Peter Jonsson is partially supported by the Swedish Research Council (VR) under grant 2021-04371.

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
