# OpenReview forum: "New Complexity Results for Structurally Restricted Numeric Planning"
_icaps-conference.org/ICAPS/2022/Workshop/HSDIP — HSDIP 2022_

### Official Review · Reviewer_gKa7 · 2022-04-21
**Very technical, hard to follow, nevertheless, a sufficient relevant contribution**

**Confidence:** 3
**Overall Score:** Accept

**Review:**

This paper analyzes the simple numeric planning ( SNP ) problem and suggests a new complexity result. Based on the previous work by Helmert, the authors show that the decision problem is NP-hard even for exactly one numeric variable and also in PSPACE and present a pseudo-polynomial algorithm to solve such tasks. The main result of this paper is that SNP with an arbitrary number of numeric causal-graph leaf variables is decidable, and lies in PSPACE if the propositional state space has a fixed size.

The topic of the paper is relevant to the workshop, clear planning paper, an interesting approach, and well-established related work,

The structure of the paper makes sense, starting with an introduction to the problem and then adding the restrictions to finally omit them and present the main complexity result. This paper is very technical, and the results seem to be sound, however, I found the paper hard to read and hard to follow, in particular the second section-- Numeric Planning. The authors used nontrivial notations quite frequently, which harms the flow of the paper. In my opinion, it could be easier to understand the paper if the authors would have used the known notations from Helmert (2002). The rest of the paper is better written but still has too many technical details and too few natural-language textual explanations.

---

### Official Review · Reviewer_WaJd · 2022-04-24
**Highly theoretical; quite relevant**

**Confidence:** 2
**Overall Score:** Accept

**Review:**

This paper looks at pushing the known theoretical results in the numeric planning space. By incrementally building on the complexity of the studied setting, the work culminates in showing that a restricted class of numeric problems (k-PENLRT) is decidable and in PSPACE.

The topic is clearly relevant to the workshop, and it does seem to make a significant contribution to our understanding of numeric planning. The technical details of the proofs are a fair bit beyond my comfort zone at times, but the general intuitions come through.

The largest question that I'm left with after reading is what the practical setting looks like for this work. In particular, a brief analysis of existing domains would be informative (are there any with the desired structure?). Further, even a conjecture as to what approximations may come from this work would be useful to include. E.g., do the results presented lend themselves to concrete strategies for solving approximations of existing numeric domains (either as a heuristic or as a semi-decidable approach)?

Ultimately, I think this paper would be welcome at HSDIP and recommend accept.

Minor correction:
>  How to define a CG in this cases remains an open question.
- Should be `case`

---

### Author Response · Authors · 2022-04-29
**Response to the Reviews**

Thank you for your hard work!

Reviewer WaJd (R1): do the results presented lend themselves to concrete strategies for solving approximations of existing numeric domains (either as a heuristic or as a semi-decidable approach)?

This is indeed an interesting topic for future work. In classical planning, causal graphs are used in multiple ways to reduce the search effort, e.g., the causal graph heuristic by Helmert, fork decomposition heuristic by Katz & Domshlak, decoupled search by Gnad et al., etc. We hope that our work is one of the building blocks that will allow the extension of these methods for simple numeric planning.